# Modafinil Administration to Preadolescent Rat Impairs Non-Selective Attention, Frontal Cortex D_2_ Expression and Mesolimbic GABA Levels

**DOI:** 10.3390/ijms23126602

**Published:** 2022-06-13

**Authors:** Valeska Cid-Jofré, Macarena Moreno, Ramón Sotomayor-Zárate, Gonzalo Cruz, Georgina M. Renard

**Affiliations:** 1Centro de Investigación Biomédica y Aplicada (CIBAP), Escuela de Medicina, Facultad de Ciencias Médicas, Universidad de Santiago de Chile, Obispo Umaña 050, Estación Central, Santiago 9160019, Chile; valeska.cid@usach.cl (V.C.-J.); macarena.moreno@ubo.cl (M.M.); 2Escuela de Psicología, Facultad de Ciencias Sociales, Universidad Bernardo O’Higgins, Santiago 8370993, Chile; 3Laboratorio de Neuroquímica y Neurofarmacología, Centro de Neurobiología y Fisiopatología Integrativa (CENFI), Instituto de Fisiología, Facultad de Ciencias, Universidad de Valparaíso, Av. Gran Bretaña 1111, Playa Ancha, Valparaíso 2360102, Chile; ramon.sotomayor@uv.cl; 4Laboratorio de Alteraciones Reproductivas y Metabólicas, Centro de Neurobiología y Fisiopatología Integrativa (CENFI), Instituto de Fisiología, Facultad de Ciencias, Universidad de Valparaíso, Av. Gran Bretaña 1111, Playa Ancha, Valparaíso 2360102, Chile; gonzalo.cruz@uv.cl

**Keywords:** modafinil, ADHD, NSA, nucleus accumbens, glutamate, GABA, psychostimulants, ventral tegmental area, prefrontal cortex

## Abstract

The misuse of psychostimulants is an increasing behavior among young people, highlighting in some countries the abuse of modafinil (MOD) as a neuropotentiator. However, several clinical trials are investigating MOD as an alternative pharmacological treatment for attentional deficit and hyperactivity disorder (ADHD) in children and adolescents. On the other hand, the early use of psychostimulants and the misdiagnosis rates in ADHD make it crucial to investigate the brain effects of this type of drug in young healthy individuals. The aim of this work was to evaluate the effects of chronic MOD treatment on neurochemicals (γ-aminobutyric acid and glutamate), dopamine receptor 2 (D_2_) expression and behavior (non-selective attention “NSA”) in the mesocorticolimbic system of young healthy Sprague–Dawley rats. Preadolescent male rats were injected with MOD (75 mg/kg, i.p.) or a vehicle for 14 days (from postnatal day 22 to 35). At postnatal day 36, we measured the GLU and GABA contents and their extracellular levels in the nucleus accumbens (NAc). In addition, the GLU and GABA contents were measured in the ventral tegmental area (VTA) and D_2_ protein levels in the prefrontal cortex (PFC). Chronic use of MOD during adolescence induces behavioral and neurochemical changes associated with the mesocorticolimbic system, such as a reduction in PFC D_2_ expression, VTA GABA levels and NSA. These results contribute to the understanding of the neurological effects of chronic MOD use on a young healthy brain.

## 1. Introduction

Modafinil (MOD) is a wakefulness-promoting drug and an atypical psychostimulant that is commonly prescribed for narcolepsy, obstructive sleep apnea/hypopnea syndrome, and shift-work sleep disorder [1,2]. MOD is also used to enhance attention and vigilance in adults [3,4]. Clinical trials are investigating whether MOD increases attention in children and adolescents diagnosed with attentional deficit and hyperactivity disorder (ADHD) [5,6,7]. However, it is described that ADHD is frequently over-diagnosed [4], therefore the treatment with psychostimulants could be incorrectly prescribed to healthy children and adolescents, since there is a lack of research on the long-term effects of MOD on young healthy populations. 

The improvement of ADHD symptoms using psychostimulants is related to dopamine (DA) neurons within the mesocorticolimbic circuitry [8,9,10]. MOD blocks the dopamine transporter (DAT) [11,12], but with a lower affinity for the DAT [13,14] than other psychostimulants such as methylphenidate (MPH) and amphetamine (AMPH), the first-line drugs for ADHD [8,9,10]. The blocking of the DAT reduces DA reuptake, thereby increasing extracellular DA levels. Nonetheless, the mechanism of action of MOD is not completely elucidated and implicates other neurotransmitter systems such as γ-aminobutyric acid (GABA) and glutamate (GLU), among others [15,16]. Evidence shows that an acute MOD administration decreases GABA release in the NAc, substantia nigra (SN), and globus pallidum (GP) but increases GLU release in the striatum [17,18,19] in adult rats. Interestingly, acute and chronic MPH fails to increase locomotor activity when glutamatergic synapses are disrupted [20,21]. Recently, our group demonstrated that chronic MOD administration in adolescent male rats decreases DA release in the NAc [22], suggesting an impairment of the mesocorticolimbic system. However, the effects of the chronic use of this drug on glutamatergic and GABAergic transmission, especially in young healthy individuals, remains to be clarified.

The mesocorticolimbic circuitry includes the ventral tegmental area (VTA), nucleus accumbens (NAc) and prefrontal cortex (PFC), and it is involved in motivated behaviors and executive functions [23,24,25]. The PFC and NAc receive dopaminergic and glutamatergic afferents from the VTA, the main dopaminergic nuclei of the circuit [26,27]. In the same way, the PFC sends glutamatergic projections to the NAc and VTA [28], and both nuclei regulate the reward, reinforcement, and locomotor-stimulating effects that are induced by psychostimulants [29,30,31,32]. Importantly, the activation of GLU and GABA receptors such as N-methyl-D-aspartic (NMDA), metabotropic GLU receptor type five (mGlu5) and GABA type B (GABA_B_) regulate the NAc and VTA DA release [33,34,35,36]. 

On the other hand, PFC DA release is essential to regulate attention, arousal, locomotor activity, and sensitization to psychostimulants [8,37], with both type 1 and 2 receptors (D_1_ and D_2_) being involved. D_1_ and D_2_ antagonists injected into the PFC impair attention [38]. Interestingly, MOD administration increases rearing behavior and decreases impulsivity in prenatal alcohol-treated rats; however, the opposite occurs in healthy normal rats [39]. Additionally, it has been shown that ADHD animal models (spontaneously hypertensive rats; SHR) have diminished non-selective attention (NSA; shorter and more frequent rearing events) compared with control rats [40,41]. Interestingly, MPH administration for 14 days increases the duration of rearing only in SHRs [40]. However, the effect of chronic MOD administration on NSA and the PFC DA system remains to be elucidated.

In summary, it is known that MOD acts directly on the DA and indirectly on the GLU and GABA systems, modifying their activity mainly in circuits linked to attention, reward, reinforcement, and locomotor activity, but the influence of early chronic MOD use on these systems is not yet fully understood. Therefore, the aim of this work was to study the effect of chronic MOD treatment on NSA and neurochemical (GLU, GABA levels and D_2_ protein levels) outcomes in healthy juvenile rats.

## 2. Results

### 2.1. GLU and GABA Tissue Content Levels in NAc and VTA

Chronic MOD treatment decreased the VTA GABA tissue levels (sum of ranks in column A, B = 65, 26; Mann–Whitney U = 5; *p* = 0.0221, Figure 1c). GLU tissue levels were not affected by MOD treatment in the NAc (sum of ranks in column A, B = 56, 49; Mann–Whitney U = 21; *p* = 0.6807; Figure 1b) or the VTA (sum of ranks in column A, B = 58, 47; Mann–Whitney U = 19; *p* = 0.535; Figure 1d), and the same result was observed regarding GABA in the NAc (sum of ranks in column A, B = 63, 53; Mann–Whitney U = 25; *p* = 0.7789; Figure 1a).

### 2.2. Extracellular GLU and GABA Levels in NAc

Reverse dialysis of 70 mM K^+^ in NAc did not produce differences in extracellular GLU levels between vehicle and MOD rats (data not shown). A one-way ANOVA with Tukey post-hoc tests revealed an effect of time, but not of treatment (interaction *p* = 0.9202; time *p* < 0.0001; treatment *p* = 0.8027, data not shown). Extracellular levels of GABA were higher in the vehicle group after the depolarizing stimulus compared to basal levels (*p* = 0.0020, Figure 2a). In the MOD-treated group, the response was in the same direction (*p* = 0.0228; Figure 2a). The mean extracellular basal levels of both GLU (mean of column A = 0.5579, mean of column B = 0.0977; Mann–Whitney U = 24; *p* = 0.7756, data not shown) and GABA (t = 1.212, df = 10.29; *p* = 0.2526) were not different between groups (Figure 2b).

### 2.3. Non-Selective Attention (NSA) and Rearing Frequency

Non-selective attention (NSA) and rearing frequency were assessed in a 60 min test immediately after the vehicle (VEH) or MOD injection. The NSA behavior, measured as the total time/frequency of rearing (relative time or time per rearing) increased on day 14 compared to day 1 in the treated group (mean rank 1 = 4.667, mean rank 2 = 15.67, mean rank difference = −11.0, Z = 2.809; *p* = 0.029, Figure 3). As expected, no changes were observed in the VEH-injected animals between the two time points, but we observed a great variation in the behavior of this group in comparison with the MOD-injected animals. 

Rearing frequency clearly increased at day 1 after acute MOD administration (mean rank 1 = 3.80, mean rank 2 = 11.33, mean rank difference = −7.533, Z = 1.83; *p* = 0.0003). However, on day 14, we did not find significant differences between the groups (Figure 4).

### 2.4. D_2_ Expression in PFC

We found a notorious reduction in PFC D_2_ expression in the MOD-treated group compared to the vehicle group (sum of ranks in column A, B = 52, 26; Mann–Whitney U = 5; *p* = 0.0411; Figure 5). Regarding intraindividual variability, the mean coefficient of variation (CV) calculated among the samples was 6.2% (range: 0% to 20%). The interindividual variation in the control group was a CV of 39% and in the MOD group it was 52%. The high variability of the control group is due to one data point with a low level that we decided not to eliminate. If we eliminate this value, then the differences are much more significant, and the CV is 17%. We consider that the variability of the MOD group was high because each rat had a different magnitude of response to MOD. Despite this, all animals in the MOD group had a lower D_2_ expression compared to the mean levels of controls. Thus, the results are significantly different despite the variation. This issue should be considered for future experiments.

## 3. Discussion

In previous research, we reported that chronic MOD treatment during preadolescence impairs social play behavior. This was associated to a lower extracellular DA level in the NAc in response to a depolarizing stimulus compared to the controls [22]. In the current study, we used the same protocol of 14 days of MOD treatment in preadolescent rats to elucidate if GLU and GABA levels in the mesolimbic pathway are involved in our previous results. Additionally, we studied if chronic MOD induced changes in attention and D_2_ protein levels in the PFC. Accordingly, we measured NSA behavior, which involves scanning, orienting, and detecting stimuli, since there is a broader kind of attentional behavior impairment in ADHD animal models (for review see [40,41]).

Our results showed lower GABA tissue levels in the VTA after MOD treatment. This finding suggests an increase in GABA release in the VTA. Considering that GABA interneurons in the VTA regulate the firing of DA neurons [42], this result could explain our previous result in which the extracellular NAc DA levels were decreased after a depolarizing stimulus in the group treated with MOD [22]. We also observed higher extracellular NAc GABA levels after a 70 mM K^+^ depolarizing stimulus compared to basal extracellular GABA levels in the MOD-treated group. Early in vivo microdialysis studies in the NAc showed that an acute i.p. administration of MOD decreases GABA release compared to AMPH [18]. Interestingly, when MOD is subcutaneously administered concomitantly with a GABA receptor antagonist, the increase in DA release is reverted [17,18]. In the case of GLU, the same team reported an increased release in the striatum only with high doses (300 mg/kg) [19]. Similarly, ex vivo and in vitro experiments showed that MOD failed to impair GABA and GLU synthesis or metabolism in the rat hypothalamus [43].

A disturbance in the GLU:GABA ratio may be relevant for the vigilance-enhancing properties of MOD since a decrease in GLU and an increase in GABA function are important in sleep-related behaviors. In the case of GLU release, this was increased with MOD in the brain areas related to sleep/wake regulation such as the ventromedial thalamus, the ventrolateral thalamus and the hippocampal formation [18]. This different outcome in the GLU and GABAergic systems suggests that MOD exerts a region-specific effect on both neurotransmitters, and it probably depends on the type of treatment or administration (single versus repeated). As we showed in our results, GABA tissue levels were lower in the VTA in treated rats, and based on previous data in our laboratory, DA release was lower than the vehicle (see [22]) and GLU release did not change. Therefore, repeated MOD might increase the inhibitory balance in the NAc due to an augmentation of GABAergic communication with the VTA, and this enhancement of inhibitory balance is not because of a decrease in GLU levels.

On the other hand, GABA neurotransmission has been implicated in social behaviors. Several studies suggest that a decrease in GABA release in the PFC is related to a decrease in social behavior. Specifically, the disruption of GABA signaling by injecting the GABA_A_ receptor (GABA_A_R) antagonist bicuculline into the PFC of adult male rats decreased social interaction and blocked the social preference over the non-social stimulus [44] without affecting sucrose preference (non-social reward). Moreover, in juvenile rats, the blockade of GABA_A_R in the lateral septum (LS) decreased social play behavior in both female and male animals. However, blocking ionotropic glutamate receptors in LS decreased social play only in females [44]. These results are contradictory to our previous results that showed a decrease in social play behavior after chronic MOD treatment [22] and the suggested increase in VTA GABA release. This discrepancy could be due to a different action of GABA in different nuclei. Interestingly, [45] showed that an extra synaptic GABA_A_R agonist decreases social interaction and social play only in adolescent rats. Taken together, it seems likely that GABA neurotransmission affects social behavior differently according to the brain area and the age of the animals. Importantly, whether NAc GABA release is related to social play behavior in adolescent rats requires further investigation.

Regarding non-selective attention (NSA), acute (day 1) MOD administration decreased NSA, although not statistical significantly. However, chronic MOD administration restored NSA to control levels. This could be linked to a decrease in salience by a novel stimulus. The decrease in NSA induced by the acute MOD administration could be because of the exacerbated increase in DA in the subcortical and cortical regions that are associated with executive processes such as attention and working memory [46,47]. Rearing frequency was increased after acute MOD administration, but this effect was lost after 14 days of treatment, suggesting behavioral tolerance. These results are in line with the locomotor results that we observed in a previous work [22]. 

Impulsivity, attention and working memory have been demonstrated to differ between healthy young and old animals [48,49,50]. Importantly, MPH treatment has differential effects depending on age [48,49,50] and the time of day of the administration [49]. In this line, our results, suggested that MOD could improve non-selective attention in healthy young rats after chronic exposure complemented with locomotor tolerance. Nevertheless, further studies are necessary to find out if these results will be similar in aged rats. 

Several investigations have proposed that alterations in attentional processes underlie a U-inverted behavior of dopaminergic transmission, i.e., hypo- or hyperdopaminergic states in the mesocorticolimbic pathway [51]. On the other hand, the chronic use of psychostimulants such as AMPH and MOD generates long-term plastic changes in areas related to memory and cognition [52,53]. These changes could be associated with the results obtained on day 14, where the NSA of the group treated with MOD was similar to those of the control group, since there is a compensatory mechanism to recover the homeostasis of the altered dopaminergic system in response of the prolonged exposure of MOD in several cortical and subcortical regions. Besides, we observed that the effect of the acute administration of MOD (day 1) induced an increase in rearing frequency compared to the administration of the vehicle. However, this difference was lost when MOD was chronically administered (14 days). Accordingly, in previous a work, using the same protocol [22], we showed that horizontal locomotor activity was enhanced only on days 1 and 7, but not on day 14 in the MOD group compared to the vehicle. 

Interestingly, our results also show a lower expression of D_2_ in the PFC after chronic MOD treatment, despite no changes in NSA. A decrease in D_2_ expression in the PFC could be explained as a compensatory mechanism to balance the extracellular DA levels due to the prolonged inhibition of the DAT induced by MOD. Consistently, DAT KO mice exhibit downregulation of both D_2_ and D_1_ [54,55]. On the contrary, in mice, 6 days of MOD i.p. injections increase the binding of the DAT agonist [^3^H] mazindol and decrease D_2_ binding in the PFC [56]. Evidence in mice lacking D_2_ demonstrated that D_2_ is essential for MOD arousal effects [57], and MOD induced a reduction in D_2_ activity and resulted in a higher activation of midbrain DA neurons [58]. Therefore, these results suggest that MOD acts by regulating DAT and DA receptors expression, thereby generating a tolerance to MOD in terms of behavioral response. 

## 4. Materials and Methods

### 4.1. Animals

Forty-five male Sprague–Dawley rats (21 postnatal days, PND) were obtained from the vivarium of the Pontificia Universidad Católica de Chile (UC CINBIOT Animal Facility funded by PIA CONICYT ECM-07). All animals were housed in groups of three, four or five (depending on weight) in transparent polysulphonate cages in the animal facility at the CIBAP, Universidad de Santiago de Chile. They were maintained with food and water ad libitum under a 12 h:12 h light–dark cycle (light on at 7.00 h), with controlled room temperature (21 ± 2 °C) and humidity (55 ± 5%). All procedures were in strict accordance with the guidelines published in the “NIH Guide for the Care and Use of Laboratory Animals” (8th ed) and principles presented in the “Guidelines for the Use of Animals in Neuroscience Research” by the Society for Neuroscience. Furthermore, and all procedures were approved by the Bioethical and Biosecurity Committee of the Universidad de Santiago de Chile (No. 615/2017). All efforts were made to minimize animal suffering and to reduce the number of animals used. Unrelated subjects were used to avoid confounding litter effects (each experimental group was made up of subjects from at least three litters).

### 4.2. Experimental Design

The treatment consisted of one daily intraperitoneal (i.p.) injection from PND 22 to PND 35 (14 days). All rats were randomly assigned into two groups: vehicle (VEH, rats received a vehicle i.p. injection of saline with tween 80 at 16:1, respectively) or MOD (rats received MOD i.p. injection of 75 mg/kg prepared in vehicle). MOD was generously donated by Laboratorio Saval S.A. (Renca, Santiago, Chile) and was prepared as shown previously [22,59]. Administration was performed between 15 and 5 min before the dark phase began. The NSA test was performed on the nights of PND 22 (day 1) and PND 35 (day 14) (between 19.00 and 23.00 h). One group of rats was used to measure GLU and GABA tissue content, one group was used for D_2_ expression in the PFC, one group of rats was used for the NSA behavioral test, and finally another group of rats was used for the NAc in vivo microdialysis experiments.

### 4.3. Neurochemistry in NAc and VTA

#### 4.3.1. GABA and GLU Tissue Content Levels

At PND 36, rats were anesthetized with isoflurane and decapitated with a guillotine for small animals (model 51330, Stoelting Co., Wood Dale, IL, USA) and brains were removed. The NAc (Bregma +2.28 to +1.28 mm approximately) and VTA (Bregma −6.48 to −7.48 mm approximately) were micro-dissected at 4 °C and weighed on an analytical balance. Tissues were collected in 400 μL of 0.2 M perchloric acid (PCA) and then homogenized. The resultant homogenates were centrifuged for 30 min at 12,000× *g* at 4 °C and then the supernatants were filtered (PTFE syringe Filter; 0.22 mm pore size, Qing Feng OEM). The filtrates were stored at −80 °C until further analysis for GLU and GABA.

#### 4.3.2. In Vivo Microdialysis in NAc

At PND 36, in vivo microdialysis experiments were performed using a previously described protocol [60]. Briefly, the animals were deeply anesthetized with urethane (1.5 g/kg i.p.) and were placed in a stereotaxic apparatus (Stoelting, Wood Dale, IL, USA). The body temperature of the animals was maintained at 37 °C with an electric blanket controlled by a thermostat. 

A concentric brain microdialysis probe (Microdialysis Probe, Harvard Bioscience; CMA-11, 6000 Daltons cut off, 2 mm membrane length) was implanted in the NAc using the following coordinates according to the atlas of the NAc [61]: +1.5 mm rostral to the bregma, 1.5 mm lateral to the midline, and −7.2 mm below dura mater. For juveniles, the coordinates were calibrated according to bregma lambda distance: bregma lambda distance/9 * x-coordinate (x = AP; ML; DL). The microdialysis probe was perfused with artificial cerebrospinal fluid (aCSF: NaCl 147 mM; KCl 2.7 mM; CaCl2 1.2 mM and MgCl2 0.85 mM; adjusted to pH 7.4) at a flow rate of 2 μL/min using an infusion pump (model 210 RWD, RWD Life Science Co, Ltd., Shenzhen, Guangdong, China). After a stabilization period of 90 min, three perfusion samples of 15 min each were collected in tubes containing 4 μL of 0.2 M PCA. At 30 min, the aCSF solution was changed to 15 min of 70 mM K^+^ solution. Between 45 and 90 min, aCSF was again perfused through the microdialysis probe. The collected perfusion samples were stored at −8 °C until analysis. At the end of each experiment, animals were decapitated, and brains were quickly removed and stored in 4% paraformaldehyde (PFA). Brain sections of 50 μm were stained with cresyl violet and examined in a light microscope to test the placement of the probes.

#### 4.3.3. GLU and GABA Analysis

GLU and GABA levels in the NAc and VTA were assessed under the different experimental conditions. An aliquot (20 μL) of the filtrate was injected into the HPLC–fluorometer and the determination of GLU and GABA was performed as described previously [62]. Briefly, 20 μL of the sample were mixed with 4 μL of borate buffer (pH 10.8), and then the mixture was derivatized by adding 4 μL of fluorogenic reagent (20 mg of orthophthaldehyde and 10 μL of β-mercaptoethanol in 5 mL of ethanol). Next, 90 s after derivatization, samples were injected into a HPLC system with the following configuration: isocratic pump (model PU-4180, Jasco Co., Ltd., Tokyo, Japan), a C-18 reverse phase column (Kromasil 3-4.6, Bohus, Sweden), and a fluorescence detector (model FP-4020, Jasco Co., Ltd., Tokyo, Japan). The mobile phase containing 0.1 M NaH_2_PO_4_ and 24.0% (*v*/*v*) CH_3_CN (pH adjusted to 5.7) was pumped at a flow rate of 0.8 mL/min. The retention time for glutamate was 1.8 min and for GABA was 11 min while the detection limit was 5 fmol/μL. Dialysate samples were analyzed by comparing the peak area and elution time with reference standards (ChromNAV 2.0, Jasco Co., Ltd., Tokyo, Japan).

### 4.4. Cognitive-Behavioral Test

#### 4.4.1. Non-Selective Attention (NSA) Test

After 60 min of acclimation in the test room and immediately after injection of the vehicle or MOD, vertical locomotor activity was recorded for 60 min in the experimental cage. Non-selective attention (NSA) was measured as previously described [40,41,63,64]. The neural substrate of NSA is represented by the anterior attention system, which includes areas of the PFC [60]. This system has been associated with the animal’s motivational state, together with the characteristics of the stimulus (subjective value of the reward [46]). In this context, research has shown that rearing time is associated with exploration behavior in novel situations and is an index of NSA [40]. Briefly, rats were individually placed in the experimental cages (17 cm × 47 cm × 26 cm) under dim red light immediately after receiving either vehicle or MOD injection, on days 1 and 14 of the protocol. Rats were allowed to freely explore the cage for 60 min. The entire sessions were recorded with two video cameras (LX-C202 model; Lynx Security, China). We measured rearing (when the animal stood upright on their hind limbs) frequency, total time spent performing rearing and the time spent performing rearing every 10 min, and NSA behavior (relative time = total time of rearing/total number of rearing). Videos were analyzed using ANY-Maze TM software (Stoelting Co., Wood Dale, IL, USA) by two independent researchers in a double-blind method. Test cages were wiped and cleaned with 20% ethanol solution between trials.

#### 4.4.2. D_2_ Expression in PFC

Rats were euthanized using isoflurane and decapitated immediately after the social play behavior test (data in [22]). The PFC was extracted using a brain matrix of 1 mm with micro punches of 1.0- and 1.5-mm diameter (+2.68 mm to +2.10 mm approximately); tissues were homogenized with a RIPA buffer with a protease inhibitor cocktail and were frozen until analysis.

The protocol for the Western blotting was a modified version of two previous works, [65] and [66]. Briefly, total protein extracts were prepared by homogenization in a sodium dodecyl sulphate (SDS) buffer containing 20% glycerol, 4.0% SDS, 10% mercapto-ethanol, and 0.1% bromophenol blue in 125 mM Tris–HCl adjusted to pH 6.8. The protein concentration of homogenates was determined using the Bradford protein quantification assay. Samples were denatured (5 min, 95 °C) in the buffer previously described. Next, 30 μg of protein were loaded into each lane of a 10% polyacrylamide gel. Electrophoresis was carried out at 80 V for 15 min and then 100 V for 2 h. Afterwards, proteins were electroblotted onto nitrocellulose membranes using 350 mA for 2.5 h. Membranes were incubated for 1 h at room temperature with a blocking solution containing 5% BSA in TBS-T. Then, they were incubated for 1 h at room temperature with the rabbit anti-dopamine D_2_ receptor antibody (AB5084P, Merck Millipore, Burlington, MA, USA) diluted to 1:1000 in TBS-T, followed by incubation with the secondary antibody, namely a rabbit polyclonal anti-GAPDH antibody (G9545, Sigma-Aldrich Co, LLC, St. Louis, MO, USA) (1:10,000, 1 h incubation). The antibody complexes were detected using a Goat Anti-Rabbit IgG Fc (HRP; ab97200). For detection we used the EZ-ECL Kit Enhanced Chemiluminescence Detection Kit (Biological Industries, Migdal HaEmek, Israel). Chemiluminescence was captured using the C-digit Blot Scanner (LI-COR Bioscience, Lincoln, NE, USA). Results were analyzed by measuring the pixel intensities of bands using the semi-quantification tool of the Image J (National Institutes of Health, Bethesda, MD, USA). All Western blots were performed in triplicate for each sample. The coefficient of variation was 6.2% for the PFC among the samples.

## 5. Statistical Analysis

All data were analyzed with the D’Agostino–Pearson test to assess normality. The Mann–Whitney test with Welch’s correction was used to determine significant differences in GLU and GABA tissue content in the NAc and VTA. A one-way ANOVA followed by Tukey post-hoc tests was used to determine significant differences between basal micro dialysate samples and post-70 mM K^+^ stimuli intra-NAc samples. The Kruskal–Wallis test followed by Dunn post-hoc was used to determine significant differences between groups in both time points (day 1 versus day 14) in the NSA and rearing frequency test. The significance was set at *p* < 0.05. The statistical analysis was carried out in GraphPad Prism v9.0 (GraphPad Software, San Diego, CA, USA). One rat from the MOD group was an outlier (more than 2 times S.D. + mean) in the GABA tissue content level analysis and was removed from the analysis. One rat from the vehicle group was excluded due to zero rearing frequency at day 1.

## 6. Conclusions

Our results show that the chronic use of MOD during a critical development period induces behavioral and neurochemical changes in the mesocorticolimbic circuitry, specifically altering GABA levels in the VTA, D_2_ expression in the PFC, and acute modification of NSA. These results contribute to the understanding of the neurophysiological and chemical mechanisms underlying the chronic use of MOD at critical periods of development, suggesting that prolonged use of MOD induces tolerance, modifying both behavioral and neurochemical substrates from the first day of use compared to the vehicle. Finally, it is important to continue investigating the short- and long-term effects on the brain reward circuitry after chronic MOD administration. This drug is gaining popularity among young students, and ADHD is highly misdiagnosed in children and adolescents, which is a sensitive population to be exposed to psychostimulants.

## Figures and Tables

**Figure 1 ijms-23-06602-f001:**
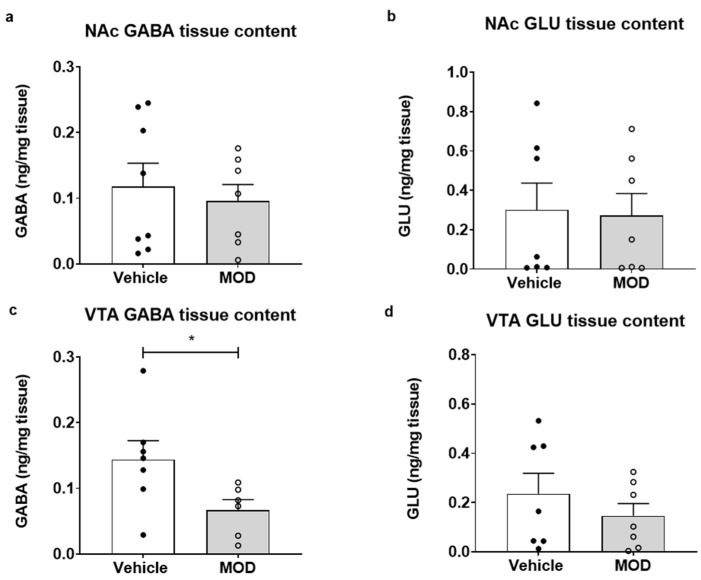
NAc and VTA glutamate (GLU) and GABA tissue content after 14 days of vehicle or MOD treatment in young rats. (**a**) NAc GABA tissue content, (**b**) NAc GLU tissue content, (**c**) VTA GABA tissue content and (**d**) VTA GLU tissue content. Data are presented as the mean ± SEM for NAc measurements: vehicle *n* = 8 and modafinil *n* = 7. For VTA measurement: *n* = 7 per group. * *p* < 0.05.

**Figure 2 ijms-23-06602-f002:**
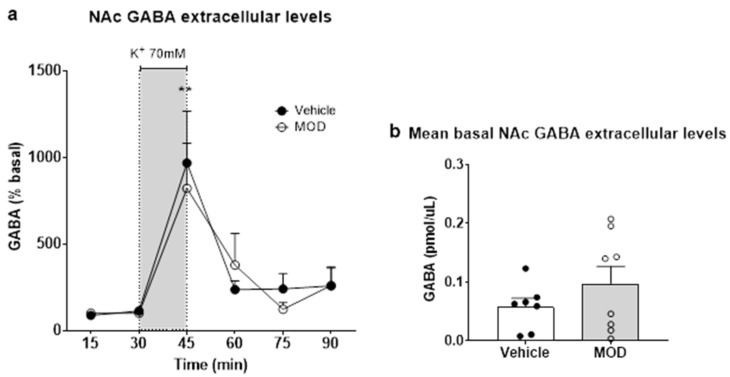
Effect of chronic MOD treatment on extracellular and release of GABA levels in the nucleus accumbens (NAc) after 70 mM K^+^ stimulation by in vivo microdialysis. (**a**) Forty-five minutes after beginning the collection of samples, an aCSF containing 70 mM K^+^ was perfused through the dialysis probe for 15 min. GABA is expressed as a percentage of baseline. Basal GABA levels (pmol/µL) for the vehicle were 0.063 ± 0.006 and vehicle K^+^ 0.476 ± 0.115; MOD basal 0.898 ± 0.413 and MOD K^+^ 1.206 ± 0.498; vehicle (*n* = 7) and MOD (*n* = 5); ** *p* < 0.01; (**b**) GABA release levels in the NAc is expressed the mean of GABA levels ± SEM.

**Figure 3 ijms-23-06602-f003:**
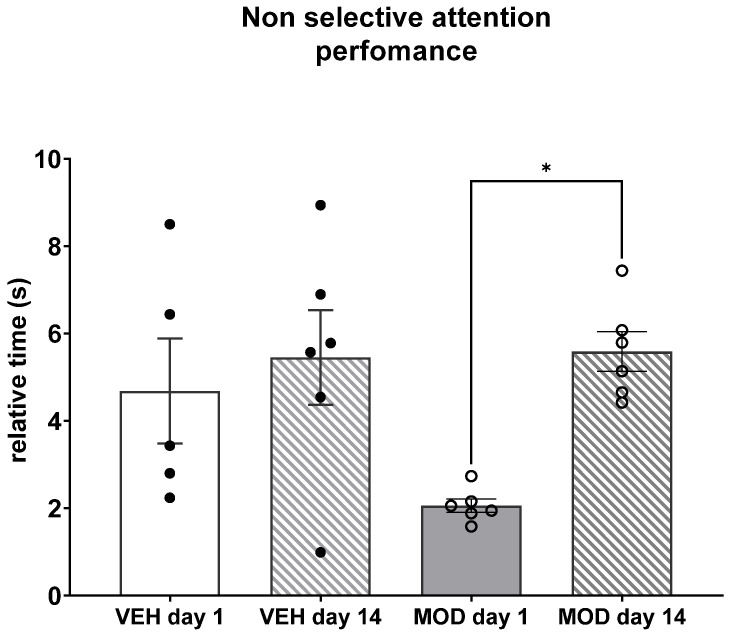
Non-selective attention (NSA) at day 1 and day 14 after vehicle (VEH) or modafinil (MOD) treatment in young male rats in 60 min test. Kruskal–Wallis test followed by Dunn post-hoc was used. Data are presented as the mean ± SEM; *n* = 5 or 6 per group. * *p* < 0.05.

**Figure 4 ijms-23-06602-f004:**
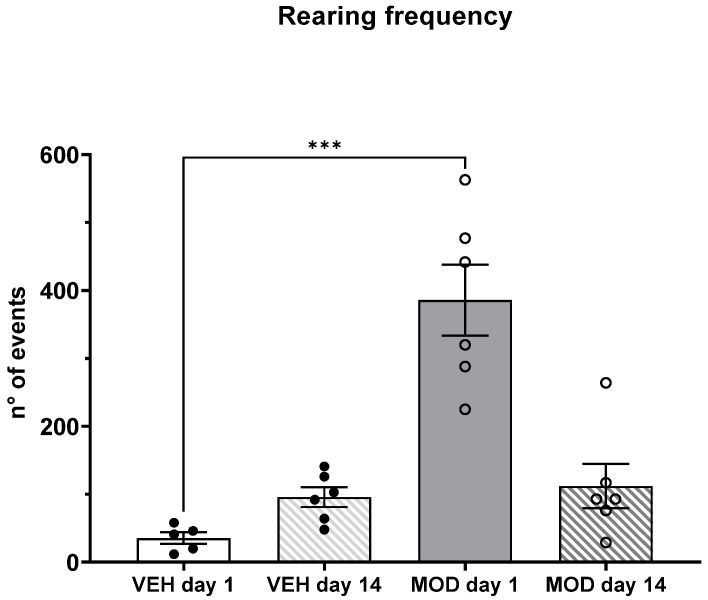
Rearing frequency at days 1 and 14 in the vehicle (VEH) and modafinil (MOD) groups of young male rats. Kruskal–Wallis test followed by Dunn post-hoc was used. Data are presented as the mean ± SEM; *n* = 5 or 6 per group. *** *p* < 0.001.

**Figure 5 ijms-23-06602-f005:**
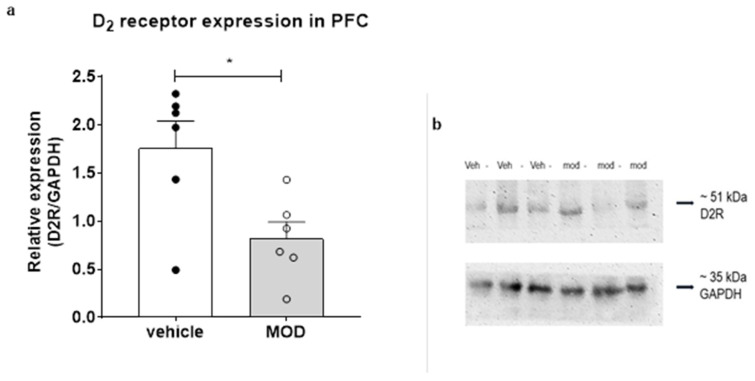
Dopamine type 2 receptor (D_2_) expression in the prefrontal cortex (PFC). (**a**) D_2_ expression in PFC (**b**) Examples of blots for expression of D_2_ in PFC after 14 days of treatment. Vehicle and MOD groups (*n* = 6), * *p* < 0.05; Data are expressed as mean ± SEM of arbitrary units of D_2_ immunoreactivity normalized to GAPDH immunoreactivity.

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
