# Peer review of "Modafinil Administration to Preadolescent Rat Impairs Non-Selective Attention, Frontal Cortex D2 Expression and Mesolimbic GABA Levels"

_ijms, 2022, doi:10.3390/ijms23126602_

Round 1

Reviewer 1 Report

The manuscript by Cid-Jofré and colleagues investigate the impact on preadolescent rats of repeated exposure of modafinil. Although potentially interesting the study in its present form is rather preliminary. Several technical issues limit the interpretation of the results.

1) Figure 1:The single values highlight the huge variability between samples questioning the way how they've been collected. Dispersion is important in vehicle treated rats suggesting that the "n" should be increased to conclude.

2) Figure 3 and 4: These two figures show the same data. Results should be presented altogether using appropriate statistical analysis. Moreover after close inspection of Figure 4 it looks like the bar graphs of panel A and B have been switched. If confirmed, rearing frequency at day 14 would also increase in modafinil treated rats.

3) Figure 5: Results of D2R expression are not convincing. Immunoblots revealed a huge variability. Moreover, in contrast to what the authors claim it is impossible to discriminate between pre and postsynaptic D2R using such approach. Additional experiments including immunofluorescence and in situ hybridization should be performed to further strengthen their observation.

Moreover, extensive editing of English language and style is required.

Author Response

Reviewer 1:

  • Figure 1: The single values highlight the huge variability between samples questioning the way how they've been collected. Dispersion is important in vehicle treated rats suggesting that the "n" should be increased to conclude.

Response: We agreed that it is an important variation between samples in vehicle group, however, the modafinil group is variable too. Therefore, we think that if we increase the “n” in both groups the results will not change.

  • Figure 3 and 4: These two figures show the same data. Results should be presented altogether using appropriate statistical analysis. Moreover, after close inspection of Figure 4 it looks like the bar graphs of panel A and B have been switched. If confirmed, rearing frequency at day 14 would also increase in modafinil treated rats.

R: Thank you very much for the suggestion. The panels A and B have not been switched; however, it could be possible that the colors of the bars used along with the data seems confusing. We done the figures again and present the results of NSA and rearing frequency altogether to avoid confusions. Also, the statistical analysis change. Please see the results section.

  • Figure 5: Results of D2R expression are not convincing. Immunoblots revealed a huge variability. Moreover, in contrast to what the authors claim it is impossible to discriminate between pre and postsynaptic D2R using such approach. Additional experiments including immunofluorescence and in situ hybridization should be performed to further strengthen their observation.

R: We appreciate the comments. Regarding variability, we had a mean coefficient of variation (CV) of 6.2 % among triplicated of the same sample (Range: 0% to 20%). Regarding interindividual variation, in the control group we got a CV of 39% while in the MOD group it was 52%. The high variability of the control group is due that one data of very low level that we decided do not eliminate. If we would eliminate this value the differences are much more significant and the CV is 17%. We consider that the variability of MOD group is high due each rat has a different magnitude of response to MOD. Despite this, as the reviewer can observe, all animals in the MOD group have a lower D2 expression compared to the mean levels of controls. Thus, the results are significantly different despite the variation.

We agree with the reviewer that immunofluorescence give a more precise location of D2 receptor (pre- or post-synaptic). We are unable to perform new experiments at this moment, therefore, we deleted the reference to pre- or post-synaptic in the text. 

  • Moreover, extensive editing of English language and style is required.

R: We revised the English language, if still you think that it is necessary, we will send to edit the English language and style.

Reviewer 2 Report

In their study, Cid-Jofre and colleagues show that chronic administration of modafinil reduces GABA tissue content in the VTA but increases basal NAc GABA extracellular levels in NAc. Locomotion is not affected by chronic treatment but increased after the first administration of modafinil. The investigation of age-specific effects of drugs is an important topic specifically when these drugs are described during development. Therefore, the topic is of great interest. However, in the current manuscript, several aspects are not clear and should be revised.

1)The introduction needs to be restructured to clarify the aim of the study

2) The method section is hard to follow.

  • Did all animals undergo the NSA test?
  • It is mentioned that animals tested for D2 were also tested for social interaction. The behavioral procedure needs to be described and the effect of this test on D2 expression should be discussed. Is it legit to compare the tissue of animals with different behavioral pre-exposure? GABA and Glutamate content might get influenced by the social test as well.
  • What is the specific attention aspect of the described behavioral test instead of just locomotion? The cited papers used a Lat-Maze which seems to differ from the described procedure. What was the rationale to modify the test?

3) The discussion should include a section on the comparison of findings in adult and preadolescent animals and discuss the developmental effects and their consequences.

Author Response

Reviewer 2:

  • The introduction needs to be restructured to clarify the aim of the study.

R: Thank you for the comment. We restructured the introduction to clarify the goal of the study. Please find the modified version in the revised version of the manuscript.

  • The method section is hard to follow.
    • Did all animals undergo the NSA test? It is mentioned that animals tested for D2 were also tested for social interaction. The behavioral procedure needs to be described and the effect of this test on D2 expression should be discussed. Is it legit to compare the tissue of animals with different behavioral pre-exposure? GABA and Glutamate content might get influenced by the social test as well.

R = We appreciate the comments. We should have been more precises about this issue. According to the suggestion we clarify this in methodology section. In brief, we used 4 different groups of animals: one to measure glutamate and GABA content; one to do microdialysis, one for D2 receptor expression (this was the only group of animals that was tested for social play behavior) and finally one group for NSA. So, we studied the effect of chronic modafinil administration only.

Regarding the effect of social test on D2 expression, the animals were euthanized just after the test and all animals were under the same protocol (vehicle or modafinil for 14 days). Therefore, we believe that D2 expression differences in the tissue between groups is due to the chronic treatment. Please find the modified version in lines: 267-270.

  • What is the specific attention aspect of the described behavioral test instead of just locomotion? The cited papers used a Lat-Maze which seems to differ from the described procedure. What was the rationale to modify the test?

R = We use this methodology for measured a broad and non-selective kind of attention processes. The non-selective attention (NSA) has been associated with the salience to reward stimuli related to the characteristics of the stimulus [9, 10]. It is focused on scanning, orienting, and detecting stimuli. In this context, time per rearing or rearing relative time is associated with exploration behavior in novel situations, and being an index of NSA [11]. We used an open field approach because their design is preferred for exploratory behavior based experiments [12]. We added this explanation at the methodology section in lines: 323-328.

  • The discussion should include a section on the comparison of findings in adult and preadolescent animals and discuss the developmental effects and their consequences.

R = Than you for the suggestion. We added this in the discussion section of the manuscript. Please find the modified version in lines: 213-219.

Round 2

Reviewer 1 Report

The authors have modified the presentation as suggested. They also provide explanation regarding the variability. I suggest to include their response to this comment in the main text (results section) to highlight this variability which should be taken into account for future experiments.

Author Response

Reviewer 1: The authors have modified the presentation as suggested. They also provide explanation regarding the variability. I suggest to include their response to this comment in the main text (results section) to highlight this variability which should be taken into account for future experiments.

R: We appreciate the comment. We added the response regarding variability in the main text (results section lines 121-130).